# Financial Regulation, Financial Inclusion and Competitiveness in the Banking Sector in SADC and SAARC Countries: The Moderating Role of Financial Stability

João Jungo *, Mara Madaleno and Anabela Botelho

Department of Economics, Management, Industrial Engineering and Tourism (DEGEIT), Research Unit on Governance, Competitiveness and Public Policies (GOVCOPP), Campus Universitário de Santiago, University of Aveiro, 3810-193 Aveiro, Portugal; maramadaleno@ua.pt (M.M.); anabela.botelho@ua.pt (A.B.)
* Correspondence: jungo@ua.pt

**Abstract:** Financial inclusion is a widely used measure to improve the living standards of households and foster inclusive economic growth. Thus, financial inclusion is one of the main policy objectives in developing countries. Besides, financial regulation (capital adequacy requirement) is a policy measure used to ensure financial stability. The objective of this study is to examine the effect of financial regulation on competitiveness and financial inclusion in 15 countries in the SADC (Southern Africa Development Community) region and 8 countries in the SAARC (South Asian Association for Regional Cooperation) region over the period 2005–2018. The result of Feasible Generalized Least Squares (FGLS) estimation suggests that financial regulation reduces competitiveness and hampers financial inclusion in the banking sector in the two regions. Furthermore, we find that financial stability moderates the negative effect of financial regulation on competitiveness and financial inclusion, meaning that financially stable banks remain competitive and normally offer financial products and services even if strong capital adequacy requirements are implemented. Additionally, we find that competitiveness increases financial inclusion in countries in the SADC region. The policy implication of this study focuses on regulatory flexibility to preserve the need for greater financial inclusion in the two regions. As for the practical implication, the study calls for strategic measures to preserve stability such as complementing financial inclusion with financial literacy, fostering corporate governance.

**Keywords:** financial regulation; financial inclusion; competitiveness; financial stability; feasible generalized least squares model

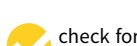



## 1. Introduction

Financial inclusion is one of the key policy objectives in the region of SADC and SAARC countries (Bara et al. 2016; Singh and Stakic 2021). From the macroeconomic point of view, the existent evidence confirms that financial inclusion contributes to fostering inclusive economic growth and effectiveness of monetary policy (Anarfo et al. 2019b; Goel and Sharma 2017; Sharmila 2019). As for the microeconomic aspect, financial inclusion is a source of poverty reduction and income inequality (Alhassan et al. 2019; Irankunda and Van Bergeijk 2020; Shinkafi et al. 2019). Financial inclusion is receiving increasing attention from policymakers for its potential to contribute to economic and financial development by promoting inclusive economic growth. However, many Asian and African economies have adopted the financial inclusion strategy as an important part of their strategies to achieve inclusive economic growth (Jayanthi and Rau 2017; Yoshino and Morgan 2017; Zhang et al. 2021). Importantly, domestic savings are the main source of financing for investment and growth in South Asia, given the limited capital flows (Sahoo and Dash 2013), for the African continent the scenario is no different (Chikalipah 2017).

Regarding the concept, financial inclusion is the elimination of any barriers that prevent households, small and medium enterprises to access and use financial products and services responsibly, safely, and offered by formal financial institutions. Financial inclusion is associated with the extension of financial services and products such as bank deposits, savings, credit, insurance, and payments (Allen et al. 2016; Huang and Zhang 2020; Ojo 2021; Yang and Zhang 2020). Importantly, increases in competitiveness in the banking sector contributed to the improved financial stability of banks (Goetz 2018). In addition, facilitating financial inclusion through cost reductions and increases in the supply of financial products and services (Albaity et al. 2019; Mengistu and Saiz 2018; Owen and Pereira 2018).

Banks play an important role in the effectiveness of financial inclusion programs and poverty reduction (Ouechtati 2020; Sikarwar et al. 2020), but for banks to perform their traditional financial intermediation functions, they must be financially stable. The 2008 financial crisis awakened policymakers to the need to implement a series of measures, such as capital adequacy, to increase the resilience of banks through better asset quality, as well as prudent risk management to avoid the accumulation of negative externalities, which can lead banks and the economy into crisis (Anarfo and Abor 2020; Gudmundsson et al. 2013; Gupta and Kashiramka 2020; Igan and Mirzaei 2020). Financial regulation (capital adequacy) is a policy instrument used to ensure the financial stability of banks and the protection of bank deposits (Gupta and Kashiramka 2020; Ugwuanyi 2015). Besides, bank recovery is more costly for the economy (Damjanovic et al. 2020). Despite the importance attributed to minimum regulatory capital requirements in banking supervision in developed and developing countries, a considerable number of banks continue to fail globally, so international financial institutions such as the International Monetary Fund (IMF) and the World Bank suggest that each country should adopt and implement the most appropriate regulatory supervision practices for its financial system (Anarfo and Abor 2020).

The expansion of bank credit to all fragments of the population through financial inclusion programs exposes financial institutions to credit risk and hence banking instability. Conversely to this, financial regulation reduces banks' ability to offer bank credit, as well as, can lead to significant increases in borrowing costs and consequently encourages the involuntary exclusion of customers who have less collateral to meet their obligations, such as the fragment of the population with less income and who need to be included in the financial system, to lend to those who need it the least or have a high income (Anarfo and Abor 2020; Igan and Mirzaei 2020). In the absence of the capital adequacy requirement, banks would be more willing to take risks to support the costs of raising capital and increase the banks' profit margin.

International financial safety nets help mitigate the adverse effects of global financial shocks and liquidity and aim to improve the domestic regulatory gap (Schuknecht and Siegerink 2020), in this regard, African and Asian central banks have been drafting laws and regulations that support the recommendations made by the Basel Committee, a fact that has enhanced financial sector stability. Therefore, full compliance with international financial regulations is contingent on the domestic needs of each country, such as financial inclusion, infrastructure financing, and financial market deepening (Goyal 2012).

Despite the importance attributed to minimum regulatory capital requirements in banking supervision in developed and developing countries, a considerable number of banks continue to fail globally, so international financial institutions such as the International Monetary Fund (IMF) and the World Bank suggest that each country should adopt and implement the most appropriate regulatory supervision practices for its financial system (Anarfo and Abor 2020).

Financial regulation, specifically the capital adequacy requirement, can force small banks with low capital to merge to comply with the regulation and maintain the normal operation. As well, it can force to close banks that are not financially stable and cause a barrier to the entry of new banks into the market (Anarfo and Abor 2020). Thus, financial regulation can be considered as a barrier to entry in the banking sector and negatively affect

competitiveness. Increasing the regulatory capital requirement concentrates the banking sector, reducing competitiveness and not ensuring the complete stability of the banking sector (Oduor et al. 2017). Opponents of consolidation argue that the pressure to increase regulatory capital is employed by large banks to prevent new entry and avoid competition (Oduor et al. 2017).

The objective of this study is to examine the effect of financial regulation on competitiveness and financial inclusion in the countries of the SADC (Southern African Development Community) and SAARC (South Asian Association for Regional Cooperation) regions, where we intend to compare the behavior of these variables in both groups of countries. To conduct the present study we extracted data in annual frequencies from different databases (widely used for data extraction in various empirical studies), specifically World Bank-World Development Indicator (WDI), Global Financial Development (GFD), and International Monetary Fund-Financial Access Survey database, for 15 SADC countries and 8 SAARC countries (See Appendix A). The study period was limited in 2005–2018 as per the largest data availability for our main variables under study (See Table 1), therefore, we used unbalanced panel data in the study. Additionally, the selection of the variables under study was based on the existing empirical literature. According to the descriptive statistics implemented in the data, we can estimate that, on average, bank regulatory capital for risk-weighted assets (regfin) in SADC and SAARC countries is 16.17% and 23.17%, respectively. For the indicator of banks' financial stability, measured by the z-score (stabfin), we find that banks in SAARC countries have an average of 15.07% and in SADC countries the average is only 9.38%. Regarding credit risk (npl), we find that on average 6.48% and 7.70% are non-performing loans in SADC and SAARC countries respectively. Comparing the credit granted by the financial sector as a percentage of Gross Domestic Product (credfs) in the two regions, we find that banks in SAARC countries granted loans with an average of 35.72% and banks in SADC countries only 27.40%. Based on these indicators, we can conclude that the financial system in SAARC countries is highly regulated and therefore more stable than the financial system in SADC countries, in addition to being more financially inclusive. Despite the progress in financial inclusion in the African and Asian continent, there is still a considerable number of populations outside the financial system.

The result of FGLS estimation shows that financial regulation negatively influences financial inclusion and competitiveness in the banking sector in SADC and SAARC regions. Furthermore, we find that financial stability moderates the negative effects of financial regulation on financial inclusion and competitiveness in the two regions under study. In other words, when banks are financially stable, the effect of financial regulation on financial inclusion and competitiveness becomes positive. Our results are robust to the use of several indicators of financial inclusion and competitiveness, as well as to the use of an aggregate sample from the two countries.

The few existing studies on financial regulation focus mainly on its effect on stability. However, some studies have analyzed the effect of financial regulation on financial inclusion. For example, Anarfo and Abor (2020) use samples from Sub-Saharan African countries, Alemu (2016) data from Ethiopia, Umoru and Osemwegie (2016) a sample from Nigeria. Authors such as Goyal (2012), Lien-Wen and Altankhuyag (2019), and Yoshino and Morgan (2017) use samples from Asian countries, whereas Gottschalk (2015) and Schaeck and Cihak (2012) employ the study in European countries. We would like to highlight that these studies do not simultaneously analyze the effect of regulation on financial inclusion and competitiveness. Moreover, we are unaware of any empirical study applied to samples from SADC and SAARC countries. Thus, this study contributes to the economic and financial literature by highlighting the effect of financial regulation in the SADC and SAARC regions, using a comparative approach between the two regions. This study not only contributes to the scarce literature on financial regulation, but also allows policymakers designing policies aimed at bank stability and promoting financial inclusion programs to know the cause-and-effect relationship between these variables (financial

regulation, financial inclusion, and competitiveness) to better manipulate and implement such measures.

The choice of the two regions is justified by the equality in economic, social, and demographic characteristics. Specifically, still visible is a large number of the adult population outside the financial system, high poverty rate, strong population growth, and strong heterogeneity in economic growth (Allen et al. 2014; Bara et al. 2016; Singh and Stakic 2021). In addition, the financial system of the two regions is still completely dependent on banks. The policy implication of this study is based on the creation of strategic measures to preserve the financial stability of banks without stifling competitiveness in the banking industry and financial inclusion. Regarding the practical and social implications, this study calls for flexibility in financial regulations so that banks can foster financial inclusion, as well as the need to adopt good corporate governance measures and complement financial inclusion with financial literacy to ensure financial stability. Furthermore, the study calls for a balance in the use of financial regulation, so as not to penalize other relevant policy objectives (financial inclusion and competitiveness) and to contribute considerably to the sustainability of economic growth, presently, common problems in the regions studied. The rest of the article is structured as follows. Section 2 provides a literature review on the subject, highlighting the contributions the present article presents to the existent literature. In Section 3 we have the presentation of the methodology, where we describe the data and specify the model. In the following Section 4, we present and discuss the results, and finally, in Section 5, we present the conclusions of the study, policy directions, limitations, and future research.

## 2. Literature Review

Policymakers responsible for creating financial regulations focus primarily on promoting financial stability, almost always ignoring the other important policy objectives such as financial inclusion and poverty reduction, competitiveness in the banking sector, inclusive economic growth (Jones and Knaack 2019; Musau et al. 2018a).

The way banks are regulated can affect their operation and their ability to provide services (Alemu 2016; Anarfo and Abor 2020; Demirguc-Kunt et al. 2003). However, when banks are forced to maintain high levels of capital, they tend to be quite cautious in offering credit and involuntarily exclude some customers due to weak credit supply and high-interest rates. This means that increases in capital requirements reduce bank lending and can promote involuntary financial exclusion (Musau et al. 2018a). Under this approach, Bridges et al. (2014) use a sample of UK banks, and later Anarfo and Abor (2020) employ the study in Sub-Saharan African countries. Both evidenced that regulation, specifically the capital requirement, reduces the ability to provide bank credit and thus makes financial inclusion programs less effective. Ugwuanyi (2015) found that strict financial regulation significantly reduces risk appetite. On the other hand, Gao and Fan (2020) showed that the implementation of macroprudential regulation increased the stability of Chinese banks. Similar results were found by Lien-Wen and Altankhuyag (2019), when they gauged that more stringent capital requirements reduced agency problems and increased banking efficiency in a sample consisting of banks belonging to the countries such as India, Thailand, Bangladesh, Malaysia, and Mongolia. In this way, we define the following research hypothesis.

**Hypothesis H1:** *Strict financial regulation may affix financial inclusion programs in the SADC and SAARC countries.*

A developed financial system provides a greater and better alternative to access financial products and services. In this perspective, Abdmoulah (2021) investigated the relationship between competitiveness and financial development and found that greater competition contributes to financial development. Within this approach, Albaity et al. (2019), and Owen and Pereira (2018) refer in their studies that greater competitiveness in

the banking sector reduces the cost of financing and increases the availability of financial products and services. This supports the studies conducted by Boyd and De Nicolo (2005), and that of Claessens and Laeven (2003), when they showed that competitiveness increases financial inclusion. Regarding the effect of financial regulation on competitiveness, Gudmundsson et al. (2013) found that regulatory efficiency increases the competitiveness of banks in Kenya. Schaeck and Cihak (2012) evidenced that rigidities in regulation decrease the competitiveness of European banks. Meanwhile, Oduor et al. (2017) found that high capital requirements reduce the competitiveness of African banks. Batuo et al. (2018) support the evidence that greater financial liberalization increases competitiveness in African countries. Thus, the second hypothesis of the present study is as follows.

**Hypothesis H2:** *Strict financial regulation reduces competitiveness in the banking sector of the SADC and SAARC countries.*

Atellu et al. (2021) studied the effect of macro and micro-prudential banking regulation on the banking sector in Kenya and found that macroprudential and micro-prudential regulation are relevant factors in achieving financial stability. Contrarily, Gaganis et al. (2021) found that macroprudential regulation decreases bank efficiency and profits.

Several empirical pieces of evidence substantiate the positive effect of financial inclusion on financial stability. Ozili (2021b) argues that financial inclusion significantly reduces financial system risk. Ahamed et al. (2021) report that financial inclusion is a source of obtaining financial resources at a considerably low cost to banks by expanding deposits. Ozili (2021a) found that financial inclusion influences and are influenced by the regulatory framework of the banking sector. From the scarce literature presented we can conclude that financial regulation can produce ambiguous results on banking efficiency. Besides, although studies of the effect of regulation on financial inclusion are increasing, these studies ignore the possibility of financial regulation, specifically suitability requirements stifling competitiveness. Moreover, we are unaware of any empirical study applied to the sample of SAADC and SAARC countries.

## 3. Methodology

### 3.1. Description of the Data

We use the World Bank World Development Indicators (WDI), Global Financial Development (GFD), and International Monetary Fund, Financial Access Statistics (IMF-FAS-Financial Access Survey) databases to obtain data for our study. These data sources are widely used in several scientific studies for the African and Asian regions (Anarfo et al. 2019a, 2019b; Anarfo and Abor 2020; Batuo et al. 2018; Oino 2015). Our sample consists of 15 SADC and 8 SAARC countries (see Appendix A), adding a full sample of 23 countries for the period 2005–2018, using annual data.

The main variables of interest in our study are financial regulation, financial stability, and financial inclusion and competitiveness indicators in the banking sector such as the Boone and Lerner index. The remaining explanatory variables used in our study are financial stability (z-score), credit risk, return on equity, financial autonomy, asset origin, bank spread, as well as macroeconomic variables such as the economic growth rate and inflation.

The z-score used as a proxy for financial stability indicates the probability of a bank failing. This measure relates the buffer of a banking system (capitalization and return) to the volatility of returns (Anarfo and Abor 2020; Feghali et al. 2021; Klapper and Lusardi 2020). To assess how financial stability moderates the effect of regulation on financial inclusion and competitiveness we create a new interactive variable composed of the variables financial stability (z-score) and financial regulation (bank regulatory capital for risk-weighted assets). It is expected that when banks are financially stable capital adequacy requirements do not affect their ability to offer financial services and products (Anarfo and Abor 2020).

The moderator variable in this context works as a mixed-mode variable, being both an independent variable and a moderator.

The use of a single measure to represent financial inclusion as the number of bank branches per thousand adults, numbers of ATMs per thousand adults, total credit granted to small-scale agriculture, and the ownership of a bank account, as implemented in the study by Grohmann et al. (2018) and Kodongo (2018) does not respect the multidimensionality of financial inclusion, as financial inclusion includes the supply and demand aspects of financial products and services. According to (Anarfo et al. 2019b; Anarfo and Abor 2020), the findings from these studies using single financial inclusion measures may be misleading and spurious and may not be relevant for policy formulation.

To respect the multidimensionality observed in the financial inclusion indicators, we also built the financial inclusion index, using principal component analysis (PCA), which aggregated the supply and demand-side dimensions of financial products and services, such as commercial bank branches per 1000 km$^2$, commercial bank branches per 100,000 adults, ATMs per 1000 km$^2$ and 100,000 adults, number of depositors in commercial banks per 100,000 adults, total bank deposits as a percentage (%) of Gross Domestic Product (GDP), total bank credit granted by the financial sector as a % of GDP, and number of borrowers in commercial banks per 100,000 adults. We follow the same approach as suggested in the study by Anarfo et al. (2019b).

Competitiveness measures in the banking sector used in several empirical studies by Albaity et al. (2019) and Goetz (2018) are the concentration of the five largest banks (C5), Panzar-Rosse H Statistics, Lerner Index, and Boone Index. In our study, to measure competitiveness in the banking sector we used the Boone and Lerner indices, as they presented the largest amount of available data.

Negative values of the Boone indicator express the existence of competition, therefore, the higher the absolute value of this indicator the stronger the competition, since this indicator evaluates competition through the efficiency channel, relating the elasticity of profits and marginal costs (Boone 2008).

The Lerner index reflects the difference between the price of products and their marginal costs and captures competitiveness, as banks with greater market power become able to set higher prices concerning costs (Lerner 1934). This index relates the elasticity and price margin that maximizes profit. Lerner's index has values between zero and one. If the market is facing perfect competition, price and marginal cost must be equal (P = MC), i.e., the Lerner index will be equal to zero. Additionally, an index equal to one (1) represents monopoly, and we are facing monopolistic competition when the Lerner index is in the range between zero and one (Lerner 1934).

The Basel Agreement suggests that banks' financial stability includes capital adequacy, asset quality, sound management, results and profitability, liquidity, and market risk sensitivity (Albaity et al. 2019; Musau et al. 2018a). Regarding financial stability, we use the z-score as a measure, which illustrates the probability of a bank failing, and can be interpreted as the number of standard deviations where profits may fall before a bank fails (Albaity et al. 2019; Anarfo and Abor 2020; Berger et al. 2009; Goetz 2018; Klapper and Lusardi 2020; Machdar 2020). Credit risk is measured by unproductive credit over the total credit portfolio of banks. This measure was also employed in studies by Albaity et al. (2019), Anarfo and Abor (2020), and Klapper and Lusardi (2020).

The banking environment is represented by the bank spread, which represents the difference between deposit rates and lending rates. However, this variable expresses the banks' profit margin and may affect banks' decisions regarding the provision of credit (Anarfo and Abor 2020; Ugwuanyi 2015). We also use returns on equity (ROE) as a control variable in our study to understand the extent to which a bank's ability to generate value from its resources and shareholders' resources can affect financial inclusion and competitiveness in the banking sector.

**Table 1.** Definition of variables and sources.

| Description | Acronym | Source |
|---|---|---|
| Financial inclusion indicator | ifi | World Bank and FMI. (WDI and FAS) |
| Electronic box per 1000 km$^2$ | atmskm | |
| Electronic box per 100,000 Adults | atmsad | |
| Seats per 1000 km$^2$ | bankm | |
| Banks per 100,000 Adults | bankad | |
| Credit granted by the financial sector in (%) of GDP | credfs | |
| Borrowers in commercial banks for 100,000 adults | borrowers | |
| Bank deposits (%) of GDP | depib | |
| Depositors in commercial banks per 100,000 adults | dbca | |
| The bank account holder for every 100,000 adults | contbank | |
| Financial regulation (regulatory banking capital for risk-weighted assets) | regfin | World Bank (WDI and GFD) |
| Financial Stability | stabfin | |
| Interaction between financial stability and regulation | regestabfin | |
| Non-performing loans/Total loans granted (Credit Risk) | npl | |
| Return on equity | roe | |
| Seat size | size | |
| Foreign assets among total assets (Origin of Assets) | assetsforeign | |
| Banking Spread | spread | |
| Boone index | iboone | |
| Lerner index | ilerner | |
| GDP growth rate | GDP | |
| Inflation | Inflation | |

Source: Own elaboration.

To measure the size of the bank we use as a proxy the bank's financial autonomy, namely the bank capital to total assets ratio. In the study by Albaity et al. (2019) to measure bank size they used the natural logarithm of banks' assets as a proxy. Bank size is a determining factor in obtaining market power and managing opportunities and asset diversification, as well as contributing to better risk management (Albaity et al. 2019).

The use of macroeconomic variables such as the rate of economic growth and the rate of inflation in the study allows inferring the impact of the countries' economic conditions on the banking sector (Mengistu and Saiz 2018). From another perspective, capital holdings also depend on macroeconomic conditions, for example, inflation is inversely related to capital, and increases in the real interest rate can affect the bank's capital ratios in the event of borrower defaults.

*3.2. Model Specification*

To assess the effect of financial regulation on financial inclusion and competitiveness in the banking sector we use the feasible generalized least squares (FGLS) model, based on the fact that the results are free from the problem of autocorrelation and heteroscedasticity (Umoru and Osemwegie 2016). In the study, using the same approach, implemented by Anarfo and Abor (2020), linear models of mixed-effects were used. In the present setting, it is believed that the FGLS estimation is adequate for the data sample used. First, by admitting changes in the time-space of the data (our panel data is not balanced, existing heterogeneity in the number of observations between the variables). Second, the FGLS model produces consistent estimates, controlling heteroskedasticity and autocorrelation. For more information on the adequacy of the model to the specific data set, please consult Parks (1967). Our FGLS model is specified in Equation (1).

$$Y_{it} = X_{it}\beta_{it} + u \tag{1}$$

where $Y_{it}$ is the vector of the k dependent variables, such as financial inclusion; competitiveness in the banking sector (Boone index, Lerner index). The $X_{it}$ is the matrix of the explanatory variables of the model, namely: financial regulation, financial stability, the interaction between financial stability and regulation, credit risk, bank spread, return on equity, bank size, the origin of banks' funds, GDP growth rate and inflation rate. The $\beta'_{it}s$ are the parameters of the vector of explanatory variables and u is the vector of random errors.

According to (Anarfo et al. 2019b; Anarfo and Abor 2020), the construction technique of the financial inclusion index is specified according to Equation (2).

$$\begin{aligned} ifi_p = W_{p1}atmskm_1 + W_{p2}atmsad_2 + W_{p3}contbank_3 + W_{p4}borrowers_4 + W_{p5}depib_5 \\ + W_{p6}credfs_6 + W_{p7}bankad_7 + W_{p8}bankm_8 + W_{p9}dbca_9 \end{aligned} \tag{2}$$

where: ifi is the financial inclusion index, $W_{pk}$ represents the weights of the respective coefficients for the country $p$. The meaning of the variables making up each index is shown in Table 1.

## 4. Results and Discussions

*4.1. Descriptive Statistics*

The result of the descriptive statistics for the full sample can be found in Appendix B. Tables 2 and 3 present the descriptive statistics for the two groups of countries under study, i.e., SADC and SAARC. We can see in the same tables as for bank account ownership (contbank) that the adult population of SADC countries has a higher average than SAARC countries, i.e., on average 408.62 and 327.67 out of every 100,000 adults in SADC and SAARC countries have a bank account.

As for the expansion of bank branches (bankm), SAARC countries have a higher average than SADC countries, indicating that for every 1000 km$^2$ there are 33.14 bank branches in SAARC countries and only 12.43 bank branches in SADC countries, illustrating that this group of African countries need to invest more resources for the emergence of more bank branches and consequently reduce one of the main barriers to the effectiveness of financial inclusion programs which is the geographical distance of institutions. Similar behavior is remarkable in the average number of ATMs (atmskm) available every 1000 km$^2$, i.e., on average in the period under analysis SADC countries provide 23.15 ATMs per 1000 km$^2$, while for South Asian countries (SAARC), this figure rises to 43.40 ATMs per 1000 km$^2$.

We found that on average the banking sector in SADC and SAARC countries collected deposits (gdp) in the order of 39.92% and 44.64% of gross domestic product, respectively, and granted bank loans in the order of 27.40% and 35.72% as a percentage of GDP. Regarding the use of financial services and products, we find that on average about 456.80 and 509.59 per 100,000 adults from SADC and SAARC countries, respectively, used bank

branches to make deposits (dbca). The number of borrowers in the two regions is still quite small, i.e., about 81.08 per 100,000 adults in SADC countries have benefited from loans, and for SAARC countries only 70.84 per 100,000 adults. From the behavior of the main indicators of financial inclusion, we can conclude that the degree of financial inclusion is higher in SAARC countries than in SADC countries.

**Table 2.** Descriptive statistics for SADC (Southern Africa Development Community) countries.

|  | Count | Mean | Sd | Min | Max |
|---|---|---|---|---|---|
| bankm | 206 | 12.468 | 29.598 | 0.058 | 111.823 |
| bankad | 206 | 9.222 | 11.667 | 0.451 | 53.348 |
| contbank | 198 | 304.396 | 408.622 | 0.000 | 1956.040 |
| atmsKm | 197 | 23.154 | 54.857 | 0.015 | 228.571 |
| atmsad | 197 | 21.160 | 21.219 | 0.098 | 82.554 |
| depib | 208 | 39.918 | 41.517 | 2.452 | 342.116 |
| dbca | 189 | 456.803 | 491.732 | 0.785 | 1956.040 |
| credfs | 207 | 27.397 | 23.878 | 2.070 | 106.260 |
| borrowers | 210 | 81.076 | 85.825 | 0.054 | 318.165 |
| regfin | 201 | 16.172 | 6.289 | 1.201 | 44.476 |
| stabfin | 210 | 9.380 | 3.879 | 2.621 | 19.225 |
| regestabfin | 210 | 152.722 | 87.175 | 3.102 | 424.626 |
| npl | 204 | 6.478 | 3.836 | 1.100 | 25.836 |
| roe | 210 | 29.511 | 18.457 | −31.280 | 137.250 |
| size | 201 | 16.240 | 14.880 | 5.700 | 67.883 |
| assetsforeign | 111 | 66.333 | 26.000 | 21.000 | 100.000 |
| spread | 206 | 14.195 | 20.945 | 0.525 | 203.375 |
| iboone | 135 | -0.078 | 0.084 | −0.411 | 0.208 |
| ilerner | 80 | 0.289 | 0.091 | 0.135 | 0.474 |
| GDP | 210 | 4.594 | 3.899 | −17.669 | 19.675 |
| Inflation | 210 | 7.778 | 6.267 | −2.409 | 36.965 |

Source: Own elaboration. See Table 1 for variables acronyms description: Financial inclusion indicator (ifi), Electronic box per 1000 km$^2$ (atmskm), Electronic box per 100,000 Adults (atmsad), Seats per 1000 km$^2$ (bankm), Banks per 100,000 Adults (bankad), Credit granted by the financial sector in (%) of GDP (credfs), Borrowers in commercial banks for 100,000 adults (borrowers), Bank deposits (%) of GDP (depib), Depositors in commercial banks per 100,000 adults (dbca), The bank account holder for every 100,000 adults (contbank), Financial regulation (regulatory banking capital for risk-weighted assets) (regfin), Financial Stability (stabfin), Interaction between financial stability and regulation (regestabfin), Non-performing loans/Total loans granted (Credit Risk) (npl), Return on equity (roe), Seat size (size), Foreign assets among total assets (Origin of Assets) (assetsforeign), Banking Spread (spread), Boone index (iboone), Lerner index (ilerner), GDP growth rate (GDP), Inflation (Inflation).

The Bank Regulatory Capital for risk-weighted assets, used as a proxy for financial regulation (regfin) in our study, averages 16.17% in SADC countries and 23.17% in SAARC countries, while for credit risk (npl) the average figures for SADC and SAARC countries are 6.48% and 7.70%, respectively. This suggests that countries in the SAARC region are highly regulated compared to SADC countries. A plausible explanation is due to the high credit risk present in the SAARC region.

**Table 3.** Descriptive statistics for SAARC (South Asian Association for Regional Cooperation) countries.

|  | Count | Mean | Sd | Min | Max |
|---|---|---|---|---|---|
| bankm | 112 | 33.143 | 40.494 | 0.119 | 180.543 |
| bankad | 112 | 9.299 | 4.967 | 0.580 | 18.585 |
| contbank | 101 | 362.767 | 327.661 | 0.000 | 1225.520 |
| atmsKm | 104 | 43.390 | 79.778 | 0.012 | 436.667 |
| atmsad | 104 | 9.855 | 8.877 | 0.059 | 36.582 |
| depib | 111 | 44.641 | 15.908 | 6.393 | 82.185 |
| dbca | 102 | 509.485 | 318.926 | 36.181 | 1225.523 |
| credfs | 112 | 35.723 | 18.443 | 3.145 | 87.967 |
| borrowers | 109 | 70.838 | 59.679 | 2.900 | 230.939 |
| regfin | 112 | 23.167 | 16.674 | 3.590 | 64.749 |
| stabfin | 112 | 15.075 | 6.724 | 4.162 | 33.407 |
| regestabfin | 112 | 353.306 | 351.718 | 41.287 | 1582.245 |
| npl | 107 | 7.697 | 6.121 | 1.100 | 49.901 |
| roe | 112 | 21.688 | 13.616 | −17.750 | 64.191 |
| size | 98 | 13.206 | 10.798 | 5.418 | 73.183 |
| assetsforeign | 45 | 18.133 | 17.101 | 2.000 | 52.000 |
| spread | 96 | 7.624 | 4.325 | 0.340 | 18.135 |
| iboone | 70 | −0.070 | 0.041 | −0.163 | 0.027 |
| ilerner | 55 | 0.228 | 0.086 | 0.003 | 0.443 |
| GDP | 112 | 5.965 | 4.278 | −13.129 | 26.111 |
| Inflation | 112 | 7.053 | 4.367 | −6.811 | 26.419 |

Source: Own elaboration. See Table 1 for variables acronyms description: Financial inclusion indicator (ifi), Electronic box per 1000 km$^2$ (atmskm), Electronic box per 100,000 Adults (atmsad), Seats per 1000 km$^2$ (bankm), Banks per 100,000 Adults (bankad), Credit granted by the financial sector in (%) of GDP (credfs), Borrowers in commercial banks for 100,000 adults (borrowers), Bank deposits (%) of GDP (depib), Depositors in commercial banks per 100,000 adults (dbca), The bank account holder for every 100,000 adults (contbank), Financial regulation (regulatory banking capital for risk-weighted assets) (regfin), Financial Stability (stabfin), Interaction between financial stability and regulation (regestabfin), Non-performing loans/Total loans granted (Credit Risk) (npl), Return on equity (roe), Seat size (size), Foreign assets among total assets (Origin of Assets) (assetsforeign), Banking Spread (spread), Boone index (iboone), Lerner index (ilerner), GDP growth rate (GDP), Inflation (Inflation).

The z-score of banks used as a proxy for financial stability (stabfin) averages 9.38% in SADC countries and 15.07% in SAARC countries. It certainly shows that banks in SAARC countries are highly regulated and are also more stable than banks in SADC countries. The bank spread represents the difference between the interest rate that banks charge on loans and the interest rate they pay on deposits. This indicator averages 14.20% for the SADC region and 7.67% for the SAARC region. However, it is notable that the bank spread (loan rate minus deposit rate) is higher in SADC countries, indicating that bank lending costs in these countries are higher compared to SAARC countries.

As for the behavior of the banking market structure in the two regions, the Boone indicator (iboone) measuring competitiveness through the efficiency channel showed average figures of −0.078 for SADC and −0.070 for SAARC indicating that there is not much difference in the degree of competitiveness in the banking markets of the two regions. The Lerner indicator (ilerner) shows averages of 0.29 and 0.23 for SADC and SAARC countries respectively, meaning that the banking market in the two regions operates in monopolistic competition.

*4.2. Unit Root Tests*

Panel data analysis requires the data to be stationary, however, the stationarity test is fundamental mainly because it allows us to avoid spurious regressions (Anarfo et al. 2019b; Anarfo and Abor 2020). To estimate the stationarity, we use Fisher-Dickey Fuller Augmented (F-ADF) and Fisher-Phillip Perron (F-PP) panel data tests, because these tests allow us to estimate the unit root for unbalanced panels. In carrying out the tests, we divided our sample into three, the first (full sample) consists of the aggregation of data from the two regions (SAADC and SAARC countries), while the second and third samples

consist of the composition of the countries of each region and were analyzed individually. Table 4 presents the result of our main variables under study, while Table 5 presents the result of the remaining control variables included in the study.

**Table 4.** Panel unit root test for the main variables of the study (endogenous and exogenous).

| | ifi | Regfin | Stabfin | Iboone | Ilerner | Regestabfin |
|---|---|---|---|---|---|---|
| **Complete Sample** | | | | | | |
| F-ADF | −0.766 | 0.304 | 4450 *** | 0.640 | 8245 *** | 0.782 |
| F-PP | 2565 ** | 0.116 | 4920 *** | 15,947 *** | 12,901 *** | 3200 *** |
| First-Dif. | | | | | | |
| F-ADF | 22,611 *** | 9471 *** | | 2750 *** | | −11,948 *** |
| PP | | 14,682 *** | | | | |
| **Sample of SADC countries** | | | | | | |
| ADF | 0.290 | 0.431 | 20,018 ** | −0.423 | 10,145 *** | 0.911 |
| F-PP | 8289 *** | −0.110 | 5234 *** | 20,208 *** | 15,775 *** | 4122 *** |
| First-Dif. | | | | | | |
| F-ADF | 2763 ** | 8462 *** | | 4097 *** | | 10,574 *** |
| F-PP | | 11,990 *** | | | | |
| **Sample of SAARC countries** | | | | | | |
| F-ADF | −2159 | −0.015 | 4865 *** | 0.814 | 0.367 | −0.509 |
| F-PP | −0.982 | 0.002 | 1172 | −0.373 | 0.702 | 0.079 |
| First-Dif. | | | | | | |
| F-ADF | −1076 *** | 2243 ** | | −0.562 *** | −0.024 *** | 3637 *** |
| F-PP | 3853 *** | 6650 *** | 18,740 *** | 5645 *** | 3300 *** | 11,388 *** |

Source: Own elaboration, ***, **, significance levels of 1% and 5%, respectively. See Table 1 for variables acronyms description: Financial inclusion indicator (ifi), Financial regulation (regulatory banking capital for risk-weighted assets) (regfin), Financial Stability (stabfin), Interaction between financial stability and regulation (regestabfin), Boone index (iboone), Lerner index (ilerner). The ifi variable was computed following Equation (2) using principal component analysis.

The results of the unit root test for the full sample and the individual sample of SADC countries indicate that the variables financial inclusion (ifi), financial regulation (regfin), Boone index (iboone), banking concentration (concentrab), the interaction between financial regulation and stability (regestabfin) and bank size (size), are integrated of order one I(1), while the remaining variables are integrated of order zero I(0).

For the SAARC country sample, the results suggest that the variables financial inclusion (ifi), Boone index (iboone), Lerner index (ilerner) are integrated of order two I(2), and the variables financial regulation (regfin), financial stability (stabfin), bank concentration (concentrab), the interaction between regulation and financial stability (regestabfin), bank size (size), asset origin (assetsforeign) and the inflation rate, are integrated of order one I(1), while the remaining variables are integrated of order zero I(0). These results suggest that not all variables included in the study follow a unit root process, however, the first differences were used for variables that are integrated of order one I(1), being second differences used for variables I(2).

**Table 5.** Panel unit root test for the control variables of the study.

| | npl | roe | Size | Assetsforeign | Spread | GDP | Inflation |
|---|---|---|---|---|---|---|---|
| **Complete Sample** | | | | | | | |
| F-ADF | 3.284 *** | 4.756 ** | −0.276 | 9.674 *** | 17.003 *** | 5.950 *** | 4.066 *** |
| F-PP | 13.739 *** | 4.779 *** | 7.861 *** | 13.514 *** | 3.857 *** | 11.273 *** | 8.235 *** |
| **First. Dif.** | | | | | | | |
| F-ADF | | | 8.153 *** | | | | |
| F-PP | | | | | | | |
| **Sample of SADC countries** | | | | | | | |
| F-ADF | 2.835 ** | 4.103 *** | −1.155 | 10.772 *** | 11.195 *** | 3.953 *** | 4.864 *** |
| F-PP | 1.322 * | 3.615 *** | 8.933 *** | 6.140 *** | 4.490 *** | 6.042 *** | 7.932 *** |
| **First. Dif.** | | | | | | | |
| F-ADF | | | 6.641 *** | | | | |
| F-PP | | | | | | | |
| **Sample of SAARC countries** | | | | | | | |
| F-ADF | 3.787 *** | 2.445 *** | 1.201 | 0.986 | 13.756 *** | 4.675 *** | 0.235 |
| F-PP | 20.409 *** | 3.153 *** | 0.860 | 15.742 *** | 0.268 *** | 10.842 *** | 3.102 *** |
| **First. Dif.** | | | | | | | |
| F-ADF | | | 4.731 *** | 4.647 *** | | | 13.515 *** |
| F-PP | | | 12.549 *** | | | | |

Source: Own elaboration, ***, **, * significance levels of 1%, 5% and 10%, respectively. See Table 1 for variables acronyms description: Non-performing loans/Total loans granted (Credit Risk) (npl), Return on equity (roe), Seat size (size), Foreign assets among total assets (Origin of Assets) (assetsforeign), Banking Spread (spread), GDP growth rate (GDP), Inflation (Inflation).

### 4.3. Impact of Financial Regulation on Financial Inclusion

The results of the FGLS estimation are presented in Tables 6–8. We use as dependent variable the financial inclusion index created through the principal component analysis and to measure the robustness of the results we expand the analysis on individual financial inclusion indicators such as bank deposits as a percentage of GDP (depib), the total number of commercial banks per 100,000 adults (bankad), and credit granted by the financial sector as a percentage of GDP (credfs).

Our main explanatory variable of interest is financial regulation, which was measured by the proxy bank regulatory capital for risk-weighted assets. In addition to this variable, we use as a control variable banking variables as an interaction between banking stability and banking regulation, the banking spread, competitiveness in the banking sector, and credit risk, as used in the study by Anarfo and Abor (2020).

Our results confirm the negative and significant impact of financial regulation on financial inclusion in SADC and SAARC countries (Tables 6 and 7). Results suggest that increases in financial regulation (expansion of bank regulatory capital into risk-weighted assets) significantly reduce banks' ability to provide financial services and products. Our results on the negative relationship between financial regulation and financial inclusion are in line with the most recent empirical findings presented in the study by Anarfo and Abor (2020), Gao and Fan (2020), Gupta and Kashiramka (2020), Igan and Mirzaei (2020), Lien-Wen and Altankhuyag (2019), and Schuknecht and Siegerink (2020), who found that financial regulation, especially the capital adequacy ratio, limits banks' ability to supply, while inhibits financial inclusion programs.

When we expanded our analysis to individual indicators of financial inclusion such as total deposits as a percentage of gross domestic product (GDP), credit provided by the financial sector as a percentage of GDP (credfs), and expansion of bank branches per 100,000 adults (bankad), we found that for the sample of SADC countries and the full sample, the negative relationship between financial regulation and bank deposits as a percentage of gross domestic product (gdp), and for credit provided by the financial sector

as a percentage of GDP (credfs) prevails. Curiously we found a positive effect of financial regulation on bank branch expansion per 100,000 adults in the sample of SADC countries, while for the sample of SAARC countries the relationship is negative, turning the impact of financial regulation on this financial inclusion indicator inconclusive in comparative terms.

**Table 6.** Sample of SADC countries.

|  | ifi | Depib | Credfs | Bankad |
|---|---|---|---|---|
| regfin | −0.0513 *** | −1.987 *** | −2.406 *** | 0.473 ** |
|  | (0.008) | (0.002) | (0.000) | (0.021) |
| regestabfin | 0.00779 *** | 0.299 *** | 0.225 *** | 0.0518 *** |
|  | (0.000) | (0.000) | (0.000) | (0.000) |
| spread | −0.0180 ** | 0.0527 | −0.537 *** | −0.0534 |
|  | (0.022) | (0.690) | (0.002) | (0.212) |
| iboone | 3.549 *** | 143.8 *** | 71.16 *** | 49.34 *** |
|  | (0.000) | (0.000) | (0.000) | (0.000) |
| npl | 0.0166 | −3.361 *** | −2.881 *** | 0.120 |
|  | (0.551) | (0.000) | (0.000) | (0.674) |
| cons | −0.0765 | 52.18 *** | 60.15 *** | −3.927 |
|  | (0.797) | (0.000) | (0.000) | (0.202) |
| N | 112 | 123 | 121 | 123 |
| wch2 | 56.69 | 91.46 | 152.26 | 66.10 |
| Prob. | 0.000 | 0.000 | 0.000 | 0.000 |

Source: Own elaboration, ***, ** significance levels of 1% and 5%, respectively. See Table 1 for variables acronyms description. Results of the FGLS model (see Equation (1)) for the sample of SADC (Southern Africa Development Community) countries. The dependent variable is the financial inclusion index created through the principal component analysis (Equation (2)) and to measure the robustness of the results we expand the analysis on individual financial inclusion indicators such as bank deposits as a percentage of GDP (depib), the total number of commercial banks per 100,000 adults (bankad), and credit granted by the financial sector as a percentage of GDP (credfs).

According to Anarfo and Abor (2020) financial regulation, specifically that requiring capital adequacy, can increase the opportunity cost of banks' capital and reduce returns on net assets. So, banks are motivated to raise lending rates and lower interest on deposits. This measure results in a disincentive to use banking services such as bank deposits and loans and thus in the failure of financial inclusion programs in SADC and SAARC countries. It is no coincidence that our results show negative effects of financial regulation on individual financial inclusion indicators such as bank deposits as a percentage of GDP (gdp) and credits provided by the financial sector as a percentage of GDP (credfs) for the sample of SADC countries and the full sample (Tables 6 and 8), although the same relationship was not statistically significant for SAARC countries (Table 7).

The interaction between financial stability and financial regulation (regestabfin) positively impacts the financial inclusion index (ifi) in the SADC country sample and the full sample, as well as in the respective individual financial inclusion indicators such as bank deposits as a percentage of GDP (depib), bank credit (credfs) and bank agency expansion (bankad). As for the behavior of the interaction between financial stability and financial regulation (regestabfin) on financial inclusion indicators in SAARC countries, the results confirm that this interaction is only statistically significant for individual financial inclusion indicators and confirm the positive effect of the interaction between financial stability and financial regulation (regestabfin) on bank deposits, bank credit and bank branch expansion. This result confirms that when banks are financially stable, financial regulation (capital adequacy requirement) does not affect their ability to provide financial services and products. This result is in line with the findings of Anarfo and Abor (2020), as in the exposure found in Musau et al. (2018b), who stated that emphasizing the financial stability of banks from the perspective of regulatory pressure can increase financial institutions' profits and mitigate credit risk, and in parallel can promote involuntary financial exclusion.

Additionally, we find that competitiveness increases financial inclusion in countries in the SADC region. However, the effect of competitiveness is positive and statistically significant in all indicators of financial inclusion. This result is in line with the results found by Mengistu and Saiz (2018), and those found by Owen and Pereira (2018). For SAARC countries we do not evidence any statistical significance of the effect of competitiveness on financial inclusion.

**Table 7.** SAARC countries' sample.

| | ifi | Depib | Credfs | Bankad |
|---|---|---|---|---|
| regfin | −0.00959 *** | −0.0786 | −0.0984 | −0.190 *** |
| | (0.004) | (0.761) | (0.710) | (0.006) |
| regestabfin | 0.0000690 | 0.0608 *** | 0.0523 *** | 0.0272 *** |
| | (0.745) | (0.000) | (0.002) | (0.000) |
| spread | −0.0286 *** | 0.0783 | −0.702 * | −0.506 *** |
| | (0.000) | (0.850) | (0.098) | (0.000) |
| iboone | −0.176 | 9.844 | −19.30 | 8.403 |
| | (0.734) | (0.791) | (0.614) | (0.399) |
| npl | −0.00535 * | −0.703 *** | −0.690 *** | −0.200 *** |
| | (0.071) | (0.002) | (0.003) | (0.001) |
| cons | 0.0193 | 37.36 *** | 32.62 *** | 12.81 *** |
| | (0.819) | (0.000) | (0.000) | (0.000) |
| N | 53 | 56 | 56 | 56 |
| wch2 | 40.37 | 34.86 | 30.22 | 72.01 |
| Prob. | 0.000 | 0.000 | 0.000 | 0.000 |

Source: Own elaboration, ***, * significance levels of 1% and 10% respectively. See Table 1 for variables acronyms description. Results of the FGLS model (see Equation (1)) for the sample of SAARC (South Asian Association for Regional Cooperation) countries. The dependent variable is the financial inclusion index created through the principal component analysis (Equation (2)) and to measure the robustness of the results we expand the analysis on individual financial inclusion indicators such as bank deposits as a percentage of GDP (depib), the total number of commercial banks per 100,000 adults (bankad), and credit granted by the financial sector as a percentage of GDP (credfs).

Competitiveness in the banking sector (iboone) positively impacts financial inclusion in SAADC countries and the full sample. For SAARC countries, this variable was not statistically significant. Regarding this result on the positive impact of competitiveness on financial inclusion, it means that when the banking sector is quite competitive, financial inclusion programs are more effective, as competitiveness considerably reduces the costs of access to financial services and products. This result corroborates several studies like those of Albaity et al. (2019), Claessens and Laeven (2003), and Mengistu and Saiz (2018), using as their main argument, that increases in competitiveness in the banking sector and the search for market power lead to the allocation of existing customer portfolios and profit margins and drive banks to accept risks to increase profit margins.

**Table 8.** Complete sample.

|  | ifi | Depib | Credfs | Bankad |
|---|---|---|---|---|
| regfin | −0.0208 ** | −1.030 *** | −1.010 *** | 0.133 |
|  | (0.046) | (0.007) | (0.000) | (0.278) |
| regestabfin | 0.00261 *** | 0.152 *** | 0.111 *** | 0.0344 *** |
|  | (0.000) | (0.000) | (0.000) | (0.000) |
| spread | −0.0178 *** | −0.864 *** | −0.806 *** | −0.207 *** |
|  | (0.002) | (0.000) | (0.000) | (0.002) |
| iboone | 2.665 *** | 134.2 *** | 71.87 *** | 36.46 *** |
|  | (0.000) | (0.000) | (0.000) | (0.000) |
| npl | −0.00795 | −1.050 ** | −1.196 *** | −0.0843 |
|  | (0.476) | (0.010) | (0.000) | (0.522) |
| cons | 0.175 | 55.51 *** | 48.47 *** | 5.953 *** |
|  | (0.339) | (0.000) | (0.000) | (0.005) |
| N | 173 | 173 | 174 | 174 |
| wch2 | 41.78 | 98.10 | 118.18 | 68.53 |
| Prob. | 0.000 | 0.000 | 0.000 | 0.000 |

Source: Own elaboration, ***, ** significance levels of 1% and 5%, respectively. See Table 1 for variables acronyms description. Results of the FGLS model (see Equation (1)) for the complete sample (SADC and SAARC) of countries. The dependent variable is the financial inclusion index created through the principal component analysis (Equation (2)) and to measure the robustness of the results we expand the analysis on individual financial inclusion indicators such as bank deposits as a percentage of GDP (depib), the total number of commercial banks per 100,000 adults (bankad), and credit granted by the financial sector as a percentage of GDP (credfs).

The spread has negative effects on the financial inclusion index (ifi) and individual financial inclusion indicators such as bank deposits (dpib), bank credit (credfs), and the expansion of bank branches (bankad) in the full sample. When we analyze the behavior of this variable in SAADC and SAARC countries, we find that the spread also negatively and significantly affects the financial inclusion index, bank credit (credfs), and the expansion of bank branches. We also find that this variable has no significant effect on bank deposits. This result clearly shows that the failure of financial inclusion programs in the two regions (SADC and SAARC) are negatively conditioned by high-interest rates on loans and low-interest rates on deposits, reducing the attractiveness of making savings and applying for a bank loan. This result coincides with that found by Anarfo and Abor (2020).

Credit risk was not statistically significant for the financial inclusion index in SAADC countries and the full sample. Contrary to this, credit risk negatively and significantly impacts individual financial inclusion indicators such as bank deposits and bank credits. For SAARC countries, credit risk negatively and significantly affects the financial inclusion index, as well as individual indicators such as bank deposits (depib), bank credits (credfs), and bank branch expansion (bankad). In an environment where credit risks are quite high, such as in the SAADC and SAARC region, it is normal for banks to be afraid to foster financial inclusion programs, mainly for two main reasons: the first is that financial inclusion allows a considerable number of low-income populations access to the financial system, increasing transactions and deposits of small amounts, which do not offset costs. The second reason is related to the poor financial literacy of the newly included, thus increasing the probability of default (Jungo et al. 2021; Musau et al. 2018a). This results in the negative impact of credit risk on financial inclusion coinciding with that found by Anarfo and Abor (2020) who concluded that default increases reduce the profitability of financial institutions and their ability to provide financial products and services.

### 4.4. The Effect of Financial Regulation on Competitiveness in the Banking Sector

Competitiveness in the banking sector is important for the efficiency of the production of banking services and the improvement of the quality and supply of these services, as well as encouraging greater innovation in the banking sector (Claessens and Laeven 2003). It has been demonstrated, theoretically and empirically, that the degree of competitiveness in the financial sector can be important for the access of firms and households to financial services and the financing of investment in the economy, causing economic growth (Claessens and

Laeven 2003; Oino 2015). The other specific reason for encouraging competitiveness in the financial sector is the link between competitiveness and stability in the financial sector (Claessens and Laeven 2003). Results are presented in Table 9.

**Table 9.** Impact of financial regulation on competitiveness.

| | Full Sample | | SADC | | SAARC | |
|---|---|---|---|---|---|---|
| | iboone | ilerner | iboone | ilerner | iboone | ilerner |
| regfin | −0.000200 | −0.0211 *** | −0.00182 | −0.0210 *** | −0.00985 * | −0.0283 *** |
| | (0.969) | (0.000) | (0.790) | (0.002) | (0.061) | (0.002) |
| stabfin | −0.00285 | −0.0264 *** | 0.00431 | −0.0101 | −0.0136 * | −0.0334 *** |
| | (0.735) | (0.003) | (0.703) | (0.357) | (0.074) | (0.010) |
| regestabfin | −0.000264 | 0.00238 *** | −0.000417 | 0.00177 *** | 0.000901 * | 0.00270 *** |
| | (0.610) | (0.000) | (0.534) | (0.008) | (0.073) | (0.002) |
| npl | −0.00409 * | 0.00442 | −0.00884 *** | 0.00301 | 0.00189 | 0.00615 * |
| | (0.058) | (0.130) | (0.005) | (0.400) | (0.284) | (0.090) |
| roe | −0.000820 | 0.00259 *** | −0.000940 | 0.00126 * | −0.00133 *** | 0.00159 * |
| | (0.121) | (0.000) | (0.168) | (0.090) | (0.000) | (0.070) |
| size | −0.0115 | −0.00186 | −0.0297 | −0.0112 | 0.0865 ** | 0.143 ** |
| | (0.481) | (0.921) | (0.101) | (0.526) | (0.026) | (0.034) |
| assetsforeign | −0.000292 | 0.000506 * | 0.000187 | 0.00208 *** | −0.00186 *** | −0.00366 *** |
| | (0.285) | (0.082) | (0.699) | (0.000) | (0.000) | (0.000) |
| spread | 0.00201 ** | 0.000182 | 0.00314 *** | −0.000648 | −0.00269 | 0.00353 |
| | (0.035) | (0.865) | (0.007) | (0.574) | (0.116) | (0.221) |
| GDP | −0.00191 | 0.00672 ** | −0.000948 | 0.00273 | −0.00332 | 0.00323 |
| | (0.444) | (0.019) | (0.748) | (0.362) | (0.197) | (0.460) |
| Inflation | −0.000248 | −0.00164 | 0.00155 | 0.00476 * | −0.00121 | −0.00269 |
| | (0.870) | (0.404) | (0.411) | (0.057) | (0.216) | (0.109) |
| _cons | 0.0878 | 0.332 *** | 0.0847 | 0.210 * | −0.0154 | 0.249 ** |
| | (0.337) | (0.001) | (0.494) | (0.085) | (0.818) | (0.030) |
| N | 132 | 106 | 97 | 73 | 36 | 34 |
| wch2 | 28,61 | 54.12 | 41.70 | 63.89 | 64.19 | 118.08 |
| prob | 0.000 | 0.000 | 0.000 | 0.000 | 0.000 | 0.000 |

Source: Own elaboration, ***, **, * significance levels of 1%, 5% and 10%, respectively. See Table 1 for variables acronyms description. Results of the FGLS model (see Equation (1)) for the complete sample (SADC and SAARC) of countries and individual samples. The dependent variables are the competitiveness in the banking sector (iboone and ilerner). Financial regulation (regulatory banking capital for risk-weighted assets) (regfin), Financial Stability (stabfin), Interaction between financial stability and regulation (regestabfin), Non-performing loans/Total loans granted (Credit Risk) (npl), Return on equity (roe), Seat size (size), Foreign assets among total assets (Origin of Assets) (assetsforeign), Banking Spread (spread), Boone index (iboone), Lerner index (ilerner), GDP growth rate (GDP), Inflation (Inflation).

As the results of statistical inference are sensitive to how competitiveness is measured, in our analysis we measure the impact of financial regulation on two different indicators of competitiveness (Boone index and Lerner index) and add into the estimation five more control variables, bank size, the origin of bank capital, the return on equity, the economic growth rate and the inflation rate, as they are indicated by the literature as being able to influence the degree of competitiveness in the banking sector (Albaity et al. 2019; Claessens and Laeven 2003). Also, Schuknecht and Siegerink (2020) have shown that banks' compliance with financial regulations is strongly influenced by the size of the bank and the existence of systematically relevant global banks.

Regarding the use of macroeconomic variables such as the economic growth rate and the inflation rate in the study, this is justified by the fact that macroeconomic stability is capable of affecting banks' performance (Claessens and Laeven 2003; Mengistu and Saiz 2018). The other reason for using the economic growth rate in our study is anchored in the results found by Kumbirai and Webb (2013) when they showed that the economic and financial crisis had a negative influence on banks' returns on assets and capital.

Our results suggest that financial regulation, specifically the regulatory capital requirement for risk-weighted assets, is statistically significant and negatively related to competitiveness in the banking sector (iboone and ilerner). This result is consistent with the literature on financial regulation, which emphasizes that rigidities in regulatory capital requirements reduce banks' ability to offer financial products and services, as well as

forcing some banks that are not financially sound to close or merge with other banks, and this can cause barriers to entry into the banking sector, thus impeding competitiveness (Alemu 2016; Bridges et al. 2014; Igan and Mirzaei 2020; Oduor et al. 2017). This result also coincides with those presented by Batuo et al. (2018) and Gudmundsson et al. (2013), when they found that financial liberalization increases competitiveness in the banking sector.

When banks' main focus is on financial stability, they become more risk-averse, cautious in offering financial services and products, but as well less competitive. Our results show that financial stability reduces competitiveness among banks in SAARC countries and the full sample, while for SADC countries this variable was not statistically significant. The other argument about the negative effect of financial stability on competitiveness found in Gudmundsson et al. (2013) and Oduor et al. (2017) is that the more stable banks consider themselves too big to fail, consequently, they engage in riskier investments, making them more vulnerable to shocks than smaller banks (Gudmundsson et al. 2013; Oduor et al. 2017).

The interaction between financial stability and regulation (regestabfin) produces positive and statistically significant effects on competitiveness in SADC, SAARC, and the full sample. This result indicates that the efficiency of financial regulation on competitiveness is effective when banks are financially stable. However, this result is consistent with the result found by Anarfo and Abor (2020), which showed that the interaction between financial stability and financial regulation improves financial inclusion.

Credit risk significantly reduces the competitiveness of the banking sector in SADC countries, as it affects profitability and bank stability. Interestingly, credit risk positively affects the competitiveness of banks in SAARC countries. Theoretical and empirical literature expresses that there is an inverse relationship between credit risk and bank profits, i.e., increases in non-performing loans reduce bank profits and increase the probability of bankruptcy (Albaity et al. 2019; Goetz 2018), but this argument underlies the behavior of this variable in SADC countries and for SAARC countries a more in-depth study is needed to better understand the reasons for the positive relationship between non-performing loans and sector competitiveness.

Increases in returns on equity (roe) increase competitiveness in the banking sector in SADC and SAARC countries, when competitiveness is measured by the Lerner index, indicating that banks with greater market power set the highest prices. Regarding the positive impact of returns on equity on competitiveness, we can argue that return on bank capital is an indicator that measures bank performance that expresses a bank's efficiency in generating profits (Albaity et al. 2019), so this result means that increased bank profits improve competitiveness in the banking sector.

The size of the bank proved to be a relevant factor for competitiveness only in banks in SAARC countries, and for the rest of the sample, this variable is not statistically significant. Bank size can influence bank stability through increased market power and the opportunity to diversify assets, i.e., large banks have greater incentives to take on excess risk and secure higher earnings (Albaity et al. 2019; Schaeck and Cihak 2012).

The banking spread is a factor that drives competitiveness in the banking sector only in SADC countries. This result shows that the higher the lending rates, the more competitive SADC banks become, justified because of the high credit risk in the banking sector in these countries. Our result supports the Gottschalk (2015) approach when he said that despite financial globalization on the African continent, lending spreads on the continent are still quite high. Similarly, the result presented by Ugwuanyi (2015) proved that the spread increases the profit of those banks that take more risks. This means that as interest rates on loans in African countries are quite high, lending banks are more exposed to the risk of default and in return make more profits.

The variable foreign assets among banks' assets proved to be statistically significant on competitiveness indicators in all three samples, although it has a different effect on the SAARC sample. In the full sample and SADC sample, this variable is not statistically significant for the Boone index (iboone). Our results on the presence of foreign assets among bank assets in SADC countries are consistent with the results found by Claessens and Laeven (2003) when they showed that the presence of foreign banks and fewer restrictions on bank activity contribute to increased competitiveness. Contrary to this, for SAARC countries our results show negative effects of the presence of foreign assets on total bank assets in competitiveness indicators. These results indicate that the presence of foreign assets among total bank assets increases competitiveness in SADC countries and reduces competitiveness in SAARC countries.

Our results confirm that the inflation rate has no statistically significant effects on competitiveness in the banking sector, on the other hand, the economic growth rate has proven to be an important factor in increasing competitiveness in the banking sector, and this variable is statistically significant in the full sample. Moreover, the variables economic growth and inflation were not statistically significant in the study by Claessens and Laeven (2003).

*4.5. Endogeneity (Robustness Check)*

To address the possible existence of the endogeneity problem, we use the instrumental variable in two-stage least squares regression (IV-2SLS). We assume that financial regulation is an endogenous variable and that its relationship with financial inclusion can be influenced by each bank's profitability, financial stability, and bank size, so we use these variables as instruments. Following the logic described in Koomson et al. (2021), Lee et al. (2022), and Mroz (1987). We employed the Durbin and Wu-Hausman tests to test for endogeneity and the Sargan and Basmann tests to assess the validity of the instruments used. The results presented in the Table 10 Are For The Three Samples Used in the study, specifically the full sample consisting of the SADC and SAARC country samples, and then for each separate sample.

For the full sample the result of Durbin and Wu-Hausman tests provide significant p-value, that is, less than the 5% significance level, therefore, we reject the null hypothesis that states that financial regulation is an exogenous variable, in favor of the alternative hypothesis, financial regulation is an endogenous variable. In this case, we specify the 2SLS regression which shows more consistent estimations relative to the OLS regression. Contrary to these results, the aforementioned tests suggest that financial regulation is an exogenous variable in the SADC and SAARC country samples respectively. Additionally, the Sargan and Basmann tests suggest that the instruments used are valid. However, the results described in the OLS and 2SLS estimation for the three samples confirm that more stringent financial regulation significantly reduces financial inclusion.

**Table 10.** IV-2SLS estimation results and endogeneity tests.

| | Full Sample | | SADC | | SAARC | |
|---|---|---|---|---|---|---|
| | OLS | IV-2SLS | OLS | IV-2SLS | OLS | IV-2SLS |
| | ifi | ifi | ifi | ifi | ifi | ifi |
| regfin | −0.0211 ** | −0.0290 *** | −0.0556 *** | −0.0601 *** | −0.00457 | −0.00850 |
| | (0.049) | (0.010) | (0.008) | (0.006) | (0.763) | (0.564) |
| regestabfin | 0.00260 *** | 0.00286 *** | 0.00773 *** | 0.00804 *** | 0.00263 *** | 0.00278 *** |
| | (0.000) | (0.000) | (0.000) | (0.000) | (0.009) | (0.003) |
| spread | −0.0179 *** | −0.0191 *** | −0.0173 ** | −0.0157 * | −0.0348 | −0.0373 |
| | (0.002) | (0.001) | (0.039) | (0.084) | (0.156) | (0.103) |
| iboone | 2.625 *** | 2.572 *** | 2.766 *** | 2.715 *** | −2.499 | −2.433 |
| | (0.000) | (0.000) | (0.003) | (0.002) | (0.304) | (0.283) |
| npl | −0.00787 | −0.00761 | 0.0140 | 0.0106 | −0.0372 *** | −0.0370 *** |
| | (0.490) | (0.499) | (0.635) | (0.724) | (0.008) | (0.003) |
| _cons | 0.177 | 0.266 | −0.0504 | −0.0234 | −0.203 | −0.148 |
| | (0.343) | (0.157) | (0.874) | (0.941) | (0.603) | (0.688) |
| Obs | 172 | 171 | 111 | 109 | 53 | 53 |
| F-statistic | 7.88 *** | | 8.69 *** | | 5.38 | |
| | (0.000) | | (0.000) | | (0.000) | |
| wch2 | | 43.36 *** | | 45.08 *** | | 30.51 *** |
| | | (0.000) | | (0.000) | | (0.000) |
| Durbin chi2 | | 4.395 *** | | 0.165 | | 0.982 |
| | | (0.036) | | (0.684) | | (0.322) |
| Wu-Hausman | | 4.327 *** | | 0.155 | | 0.869 |
| | | (0.039) | | (0.695) | | (0.356) |
| Sargan | | 4.525 | | 1.179 | | 5.725 |
| | | (0.104) | | (0.555) | | (0.057) |
| Basmann | | 4.430 | | 1.104 | | 5.499 |
| | | (0.109) | | (0.576) | | (0.066) |

Source: Own elaboration, ***, **, * significance levels of 1%, 5% and 10%, respectively. See Table 1 for variables acronyms description. Results presented in this Table are for IV-2SLS estimation results and endogeneity tests. OLS results are also used for comparison purposes and robustness checks. The dependent variable is the financial inclusion index (ifi) computed by Equation (2) through principal component analysis. Results are presented for the full sample as well as for each of the individual samples: SAARC (South Asian Association for Regional Cooperation) and SADC (Southern Africa Development Community) countries. Financial inclusion indicator (ifi), Financial regulation (regulatory banking capital for risk-weighted assets) (regfin), Financial Stability (stabfin), Interaction between financial stability and regulation (regestabfin), Non-performing loans/Total loans granted (Credit Risk) (npl), Boone index (iboone).

The results of the OLS and 2SLS estimation indicate that greater competitiveness among banks contributes to increased financial inclusion. Therefore, these results mean that the need for greater access to financial products and services for the disadvantaged population easily and safely in SADC and SAARC countries is stifled by the high regulation of the banking sector; on the contrary, greater competition among banks in these economies can contribute to the effectiveness of financial inclusion programs.

## 5. Conclusions

Banks play an important role in the effectiveness of financial inclusion programs and inclusive economic growth, but for banks to perform their core financial intermediation functions, they must be financially stable. Regulation in the financial system should aim at maintaining the financial stability of financial institutions and promoting economic growth. However, combining these two objectives is quite difficult, as a strong focus on financial stability can stifle financial inclusion programs and increase the barrier to entry for new banks (Anarfo and Abor 2020; Musau et al. 2018a). A clear example of this paradox is the requirement of capital adequacy requirements that significantly reduces banks' ability to offer financial products and services, reduce risk appetite, and raise funding costs (Anarfo

and Abor 2020; Damjanovic et al. 2020; Gupta and Kashiramka 2020). Moreover, simple financial regulation does not guarantee financial stability (Anarfo and Abor 2020; Igan and Mirzaei 2020).

Given the positive effects of financial inclusion for households and the overall economy, policymakers and international bodies such as the World Bank, United Nations, Group of 20 (G-20) consider financial inclusion as a priority measure on policy agendas (Emara and El Said 2021; Pham and Doan 2020; Saha and Dutta 2020, 2022). Similarly, financial inclusion is a priority need in countries in the SADC and SAARC region (Bara et al. 2016; Singh and Stakic 2021). The results of descriptive statistics implemented on the data from 15 countries in the SADC region and 8 SAARC countries for the study period 2005–2018 suggest that on average for every 100,000 adults only 408.62 have a bank account in SADC countries and 327.67 adults have a bank account in SAARC countries. As for the average number of bank branches per 1000 km$^2$, there are only 12.43 branches in SADC countries and 33.14 in SARC countries. Therefore, access to a bank account is the first requirement for financial inclusion (Demirguc-Kunt et al. 2018). In this way, it is easily understood that financial inclusion is absent and a clear problem in the countries in the SADC and SAARC region. The purpose of this study was to examine the effect of financial regulation on competitiveness and financial inclusion in countries in the SADC region and countries in the SAARC region, as well as, to compare the results in the two regions.

The result of the FGLS estimation confirms that financial regulation causes negative and statistically significant effects on competitiveness and financial inclusion, meaning that more stringency in capital adequacy requirements can undermine financial inclusion programs and competition in the banking sector of countries in the SADC and SAARC region. Furthermore, we find that increases in competitiveness in the banking sector of SADC countries can favor financial inclusion, as evidenced in the study conducted by Mengistu and Saiz (2018) and Pham et al. (2019). For SAARC countries we can gauge nothing regarding the effect of competitiveness on financial inclusion. Additionally, our results confirm that financial stability moderates the negative effect of regulation, indicating that when the bank is financially stable, financial regulation does not reduce its competitive ability, nor its ability to offer financial products and services. The results were robust to the use of various indicators of financial inclusion and competitiveness.

Regarding the practical and social implications, this study calls for flexibility in financial regulations so that banks can foster financial inclusion, as well as the need to adopt good corporate governance measures and complement financial inclusion with financial literacy to ensure financial stability. Furthermore, the study calls for a balance in the use of financial regulation, so as not to penalize other relevant policy objectives (financial inclusion and competitiveness) and to contribute considerably to the sustainability of economic growth, which is still a problem in the regions studied. For future studies, we suggest examining the effect of the interaction between financial literacy and financial regulation on financial inclusion and competitiveness in the banking sector. Also, as more data becomes available, we suggest the inclusion of more countries in the sample, a larger period, and an appropriate comparison between continents or different regions provided the identified differences.

**Author Contributions:** J.J. was responsible for Conceptualization; Data curation; Formal analysis; Investigation; Methodology; Roles/Writing—original draft. M.M. was responsible for Investigation; Methodology; Project administration; Resources; Software; Supervision; Validation; Visualization; Roles/Writing—original draft; Writing—review & editing. A.B. was responsible for Resources; Software; Supervision; Validation; Visualization; Roles/Writing—original draft; Writing—review & editing. All authors have read and agreed to the published version of the manuscript.

**Funding:** This research received no external funding. The APC was funded by the authors.

**Data Availability Statement:** We declare that the data source is described in the text. The data support the findings of this study are available from the first author upon reasonable request.

**Acknowledgments:** The authors acknowledge the support provided by the Research Unit on Governance, Competitiveness and Public Policy (UIDB/04058/2020), funded by national funds through FCT-Fundação para a Ciência e a Tecnologia. We acknowledge the valuable suggestions of both the Editor and reviewers of this article, which let us improve it to its current version.

**Conflicts of Interest:** The authors declare no conflict of interest.

## Appendix A

**Table A1.** Countries in the Sample.

| SADC | | SAARC |
|---|---|---|
| Angola | Namibia | Afghanistan |
| Botswana | RDC | Bangladesh |
| Eswatini | Seychelles | Bhutan |
| Lesotho | South Africa | India |
| Madagascar | Tanzania | Maldives |
| Malawi | Zambia | Nepal |
| Mauritius | Zimbabwe | Pakistan |
| Mozambique | | Sri Lanka |

Source: Own elaboration. SAARC (South Asian Association for Regional Cooperation) and SADC (Southern Africa Development Community).

## Appendix B

**Table A2.** Descriptive statistics of the full sample.

| | Obs | Mean | Sd | Min | Max |
|---|---|---|---|---|---|
| bankm | 322 | 19.526 | 35.031 | 0.058 | 180.543 |
| bankad | 322 | 9.171 | 9.796 | 0.451 | 53.348 |
| contbank | 322 | 305.334 | 398.416 | 0.000 | 2019.340 |
| atmsKm | 322 | 28.031 | 63.463 | 0.000 | 436.667 |
| atmsad | 322 | 16.108 | 18.599 | 0.000 | 82.554 |
| depib | 319 | 41.368 | 34.978 | 0.000 | 342.116 |
| dbca | 322 | 391.147 | 438.060 | 0.000 | 1956.040 |
| credfs | 322 | 30.218 | 22.368 | 2.070 | 106.260 |
| borrowers | 322 | 74.948 | 77.208 | 0.000 | 318.165 |
| regfin | 313 | 18.650 | 11.413 | 1.201 | 64.749 |
| stabfin | 322 | 11.361 | 5.728 | 2.621 | 33.407 |
| regestabfin | 322 | 221.514 | 231.581 | 3.102 | 1542.526 |
| npl | 305 | 6.846 | 4.777 | 1.100 | 49.901 |
| roe | 322 | 26.790 | 17.316 | −31.280 | 137.250 |
| autofin | 299 | 15.246 | 13.733 | 5.418 | 73.183 |
| assetsforeign | 156 | 52.429 | 32.291 | 2.000 | 100.000 |
| spread | 302 | 12.106 | 17.722 | 0.340 | 203.375 |
| iboone | 199 | −0.077 | 0.072 | −0.411 | 0.208 |
| ilerner | 135 | 0.264 | 0.094 | 0.003 | 0.474 |
| GDP | 322 | 5.071 | 4.081 | −17.669 | 26.111 |
| Inflation | 322 | 7.526 | 5.682 | −6.811 | 36.965 |

Source: Own elaboration. See Table 1 for variables description: Financial inclusion indicator (ifi), Electronic box per 1000 km$^2$ (atmskm), Electronic box per 100,000 Adults (atmsad), Seats per 1000 km$^2$ (bankm), Banks per 100,000 Adults (bankad), Credit granted by the financial sector in (%) of GDP (credfs), Borrowers in commercial banks for 100,000 adults (borrowers), Bank deposits (%) of GDP (depib), Depositors in commercial banks per 100,000 adults (dbca), The bank account holder for every 100,000 adults (contbank), Financial regulation (regulatory banking capital for risk-weighted assets) (regfin), Financial Stability (stabfin), Interaction between financial stability and regulation (regestabfin), Non-performing loans/Total loans granted (Credit Risk) (npl), Return on equity (roe), Seat size (size), Foreign assets among total assets (Origin of Assets) (assetsforeign), Banking Spread (spread), Boone index (iboone), Lerner index (ilerner), GDP growth rate (GDP), Inflation (Inflation).

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
