# Peer review of "Financial Regulation, Financial Inclusion and Competitiveness in the Banking Sector in SADC and SAARC Countries: The Moderating Role of Financial Stability"

_ijfs, doi:10.3390/ijfs10010022_

Round 1
Reviewer 1 Report
Dear Authors,
After reading the first version of your manuscript, I am still not fully persuaded that the article goes into detail enough to provide an added value to IJFS readership.
This time, I will try to give my comments in a more clear version:
Authors can name the article concisely.
In the Abstract, the author needs to write more about the research sample and summarize the highlights of the research results.
The author needs to clarify the new contribution of the research in the introduction. This part the author did was too sketchy. It is necessary to clearly state the new and motivating points of the article.
The literature review does not show but only the research context. the author needs to add and hypothesis development.
Authors need to update recent studies. And point out the missing point to carry out this study. The author should have a literature review to compare the results of previous studies conducted in the same research context.
The author needs to perform a GMM regression and endogenous phenomenon test. FGLS may not fit the data.
Regarding the moderator variable, it is necessary to clarify whether it is a pure moderator or a mixed-mode variable (both an independent variable and a moderator).
Kind regards,
Reviewer
Author Response
Dear Editor,
Dear Reviewers,
We have conformed to all the requested changes and the entire article has been reviewed following the suggestions and queries raised. Below you will find the point-by-point answers and in the manuscript, all changes have been marked in blue to turn easier the reading under suggestions. These suggestions have been acknowledged in the appropriate section.
Reviewer #1
After reading the first version of your manuscript, I am still not fully persuaded that the article goes into detail enough to provide an added value to IJFS readership.
This time, I will try to give my comments in a more clear version:
Authors can name the article concisely.
Thanks for the suggestion. The title of the article has been rewritten following the suggestion to turn it more directly. Now we have named it: “Financial regulation, financial inclusion and competitiveness in the banking sector in SADC and SAARC countries: The moderating role of financial stability”
In the Abstract, the author needs to write more about the research sample and summarize the highlights of the research results.
The abstract has been rewritten following the suggestions. Thanks for calling our attention to this fact. The new abstract looks like this now:
Financial inclusion is a widely used measure to improve the living standards of households and foster inclusive economic growth. Thus, financial inclusion is one of the main policy objectives in developing countries. Besides, financial regulation (capital adequacy requirement) is a policy measure used to ensure financial stability. The objective of this study is to examine the effect of financial regulation on competitiveness and financial inclusion in 15 countries in the SADC (Southern Africa Development Community) region and 8 countries in the SAARC (South Asian Association for Regional Cooperation) region over the period 2005-2018. The result of Feasible Generalized Least Squares (FGLS) estimation suggests that financial regulation reduces competitiveness and hampers financial inclusion in the banking sector in the two regions. Furthermore, we find that financial stability moderates the negative effect of financial regulation on competitiveness and financial inclusion, meaning that financially stable banks remain competitive and normally offer financial products and services even if strong capital adequacy requirements are implemented. Additionally, we find that competitiveness increases financial inclusion in countries in the SADC region. The policy implication of this study focuses on regulatory flexibility to preserve the need for greater financial inclusion in the two regions. As for the practical implication, the study calls for strategic measures to preserve stability such as complementing financial inclusion with financial literacy, fostering corporate governance.
The author needs to clarify the new contribution of the research in the introduction. This part the author did was too sketchy. It is necessary to clearly state the new and motivating points of the article.
Thanks for the suggestion. The introduction section has also been rewritten to clearly state the new and motivating points of the article, as suggested. We advise the reading of the introduction since it is too much to be reproduced here.
The literature review does not show but only the research context. the author needs to add and hypothesis development.
Following this suggestion, and thanks in advance, we have also rewritten the literature review in section 2, adding and developing the hypothesis under study. Being a big restructured section we would like to gently ask the reviewer to follow its reading again.
Authors need to update recent studies. And point out the missing point to carry out this study.
More recent studies have been added as requested and the motivation was reinforced in the introduction, as already stated previously. Thanks.
The author should have a literature review to compare the results of previous studies conducted in the same research context.
Thanks for the valuable suggestion. We have tried to compare our results with those of previous authors and include the recent citations now employed. Although we would like to stress that we have not found any study about regulations, inclusion, and competitiveness in the context of SADC and SAARC countries, enabling the comparison.
The author needs to perform a GMM regression and endogenous phenomenon test. FGLS may not fit the data.
Thanks for the suggestion. In the present context and this has been highlighted in the manuscript in section 3, we believe that the FGLS estimation is adequate to the data sample used. First, by admitting changes in the time-space of the data (our panel data is not balanced, existing heterogeneity in the number of observations between the variables). Second, the FGLS model produces consistent estimates, controlling heteroskedasticity and autocorrelation. For more information on the adequacy of the model to the specific data set, please consult Parks, R. (1967). Efficient estimation of a system of regression equations when disturbances are both serially and contemporaneously correlated. Journal of the American Statistical Association, 62(318), 500–509. Additionally, we have already in the analysis robustness checks using different measures of the dependent variables (besides the created index) to test the validity of our results.
Regarding the moderator variable, it is necessary to clarify whether it is a pure moderator or a mixed-mode variable (both an independent variable and a moderator).
Thanks for the suggestion. The moderator variable in this context works as a mixed-mode variable, being both an independent variable and a moderator. This has been highlighted in the manuscript.

Reviewer 2 Report
Dear Authors,
I appreciate your great effort when preparing the manuscript. The article presents a complex theoretical study on the impact of financial regulation on financial inclusion and competitiveness in the banking sector. The research findings are supported by a robust quantitative model using different econometric approaches (including PCA when modeling the financial variables).
Please consider my comments as follows:
- The introductory part can better explain the relationship between competitiveness and financial inclusion in the banking sector. Please provide clear definitions of these categories and linkage between them so the reader can easier follow your model and reasoning in the other parts of the research. You can also indicate that financial inclusion might have two aspects and focuses of study: in the broader macroeconomic context as the degree of access to valuable and affordable financial products and services that meet customer’s needs, and the second, narrower context, as the instrument of poverty reduction (as you state in line 30-31 of the manuscript). According to my understanding, in the current study, you focus on the broader macroeconomic context of financial inclusion and try to describe the relationship of this phenomenon with other channels shaping the economic condition of the banking sector (including financial stability and competitiveness). Thus, I think you could clearly indicate your approach in the first part of the article.
- I believe you should pay more attention to the regional aspect of the study (Asia and Africa), probably by comparing these regions to the benchmark from the other region with a well-developed banking sector to reflect better the problem of financial inclusion and competitiveness in the countries under research (alternatively, you can try using the single country as a benchmark; maybe South Africa from the sample). Moreover, in a descriptive part of the study, you could also provide some figures reflecting the level and discrepancies in financial literacy, financial attitude, and economic behavior of the residing societies and entities in the studied regions to support your reasoning on the importance of financial inclusion in a broader context. Again, comparing communities of the studied regions (countries) to the country with a well-developed banking sector could be helpful here.
Author Response
Editor and Reviewers answers
Dear Editor,
Dear Reviewers,
We have conformed to all the requested changes and the entire article has been reviewed following the suggestions and queries raised. Below you will find the point-by-point answers and in the manuscript, all changes have been marked in blue to turn easier the reading under suggestions. These suggestions have been acknowledged in the appropriate section.
Reviewer #2
Please consider my comments as follows:
- The introductory part can better explain the relationship between competitiveness and financial inclusion in the banking sector. Please provide clear definitions of these categories and linkage between them so the reader can easier follow your model and reasoning in the other parts of the research. You can also indicate that financial inclusion might have two aspects and focuses of study: in the broader macroeconomic context as the degree of access to valuable and affordable financial products and services that meet customer’s needs, and the second, narrower context, as the instrument of poverty reduction (as you state in line 30-31 of the manuscript). According to my understanding, in the current study, you focus on the broader macroeconomic context of financial inclusion and try to describe the relationship of this phenomenon with other channels shaping the economic condition of the banking sector (including financial stability and competitiveness). Thus, I think you could clearly indicate your approach in the first part of the article.
We would like to thank the reviewer for the valuable suggestion. As also requested by another reviewer, the introduction had to be rewritten and we took all these observations into account. We please ask the reviewer to read the entire rewritten introductory section again, please, since it is too much to reproduce all here.
- I believe you should pay more attention to the regional aspect of the study (Asia and Africa), probably by comparing these regions to the benchmark from the other region with a well-developed banking sector to reflect better the problem of financial inclusion and competitiveness in the countries under research (alternatively, you can try using the single country as a benchmark; maybe South Africa from the sample). Moreover, in a descriptive part of the study, you could also provide some figures reflecting the level and discrepancies in financial literacy, financial attitude, and economic behavior of the residing societies and entities in the studied regions to support your reasoning on the importance of financial inclusion in a broader context. Again, comparing communities of the studied regions (countries) to the country with a well-developed banking sector could be helpful here.
The results discussion and conclusions have also been rewritten. An appropriate comparison as requested was included in this new version. Thanks for all the valuable suggestions.

Reviewer 3 Report
In my opinion paper, “How does financial regulation stifle financial inclusion and competitiveness in the banking sector in SADC and SAARC countries? The moderating role of financial stability”, describes a well-known problem in the literature. However, it can be still interesting for readers.
General remark.
- The literature review is well done but should be shown a scientific gap more clearly. Unfortunately, the main concept, timeliness and research problem are not clearly presented. The justification should reveal the topicality and the main problem of the research. Authors formulated research question: “What is the effect of financial regulation on financial inclusion and competitiveness in the banking sector?” The primary purpose of financial regulations is to ensure the security of the banking system. So this obviously restricts competition. (248) This analysis is not necessary.
- Data isn’t clearly explained, e.g.:
- Is the data homogeneous?
- What is the z-score ratio? How is it constructed? Why does it measure financial stability? (is it dummies? - wealth area, financial distress area and a grey area).
- Add the argument to data selection and reasoning for their representativeness and data quality.
- Model?
- The Authors analyse time series using the regression model. In the first part, the ADF model is used to check variables stationary the problems are:
- Time series is too short (only 14 years) to use ADF test (probably is not possible to test this problem, maybe KPSS test is better)
- The ADF test is not exactly a stationary test but a test for the occurrence of a unit root equal to one. It means that there are cases when we reject the hypothesis, but the process is not de facto stationary).
- In my opinion, the Authors should use the panel regression model (fixed or random effects) and choose appropriate model’s statistics (e.g. the Breusch-Pagan test, Durbin-Wu-Hausman test etc.)
- Results should be more deeply explained and described. I have not seen the limitation, advantages and disadvantages of the Authors analysis. I could not confirm that these results are valid.
I recommend to:
- The scientific problem of the paper (research) has to be corrected. The research question should be changed to convert to the hypothesis.
- Should extend their considerations because they are not clear to all potential readers
- More clearly, explain and emphasise the main concept, research problem, contribution to the science and hypothesis in comparison with existing literature
- Do new analysis and build new panel regression models (fixed or random effects) and choose appropriate model’s statistics (e.g. Breusch-Pagan test; Durbin-Wu-Hausman test etc.)
- The conclusion should be extended. (limitation, what are the advantages and disadvantages of this analysis?)
In general, if the Authors do major revisions - rebuild and complete their data analysis and aim, this study will be valuable.
Author Response
Reviewer #3
In my opinion paper, “How does financial regulation stifle financial inclusion and competitiveness in the banking sector in SADC and SAARC countries? The moderating role of financial stability”, describes a well-known problem in the literature. However, it can be still interesting for readers.
Thanks for the observation. It is a well-known problem, but not empirically tested with the relevance and focus as we try to do in the present manuscript. Contributions have been properly identified and explored in this revised version.
General remark.
- The literature review is well done but should be shown a scientific gap more clearly. Unfortunately, the main concept, timeliness and research problem are not clearly presented. The justification should reveal the topicality and the main problem of the research. Authors formulated research question: “What is the effect of financial regulation on financial inclusion and competitiveness in the banking sector?” The primary purpose of financial regulations is to ensure the security of the banking system. So this obviously restricts competition. (248) This analysis is not necessary.
Thanks for the valuable suggestions. As already requested by other reviewers the entire section of the introduction has been rewritten and we advise for its new reading, please. Moreover, clear hypotheses have been presented and justified in the literature review section.
- Data isn’t clearly explained, e.g.:
- Is the data homogeneous?
As we can see by the number of observations indicated in the data descriptive statistics, the sample is not homogeneous for all variables, a fact that is clearly stated in the manuscript as well as we have written about the unbalanced nature of the data collected and used in this study. Thanks for the observation.
- What is the z-score ratio? How is it constructed? Why does it measure financial stability? (is it dummies? - wealth area, financial distress area and a grey area).
Following the suggestion, a concrete explanation was included in the article regarding these issues, as well as the nature of the variables have been stated to turn clear.
- Add the argument to data selection and reasoning for their representativeness and data quality.
Following the suggestion, a concrete explanation and reasoning of data selection were included in the article. They are represented in the context, having in mind the data availability also exposed. Thanks for the suggestion.
- Model?
- The Authors analyse time series using the regression model. In the first part, the ADF model is used to check variables stationary the problems are:
- Time series is too short (only 14 years) to use ADF test (probably is not possible to test this problem, maybe KPSS test is better)
- The ADF test is not exactly a stationary test but a test for the occurrence of a unit root equal to one. It means that there are cases when we reject the hypothesis, but the process is not de facto stationary).
Thanks for the raised queries and suggestions. It is important to mention that the sample data is unbalanced panel data, which means that in these situations the most appropriate tests to infer about data stationarity are those of Fisher and Im-Pesaran-Shin type. The KPSS test for unit roots is more appropriate when we have time series which is not the case. To support our justifications we advise the reading of the following econometrics articles, which were followed in the present study to implement all the estimation steps followed.
Breitung, J. 2000. The local power of some unit root tests for panel data. Advances in Econometrics, Volume 15: Nonstationary Panels, Panel Cointegration, and Dynamic Panels, ed. B. H. Baltagi, 161–178. Amsterdam: JAY Press.
Breitung, J., and S. Das. 2005. Panel unit root tests under cross-sectional dependence. Statistica Neerlandica 59: 414–433.
Choi, I. 2001.Unit root tests for panel data. Journal of International Money and Finance 20: 249–272.
Hadri, K. 2000. Testing for stationarity in heterogeneous panel data. Econometrics Journal 3: 148–161.
Harris, R. D. F., and E. Tzavalis. 1999. Inference for unit roots in dynamic panels where the time dimension is fixed. Journal of Econometrics 91: 201–226.
Im, K. S., M. H. Pesaran, and Y. Shin. 2003. Testing for unit roots in heterogeneous panels. Journal of Econometrics 115: 53–74.
Levin, A., C.-F. Lin, and C.-S. J. Chu. 2002. Unit root tests in panel data: Asymptotic and finite-sample properties. Journal of Econometrics 108: 1–24.
- In my opinion, the Authors should use the panel regression model (fixed or random effects) and choose appropriate model’s statistics (e.g. the Breusch-Pagan test, Durbin-Wu-Hausman test etc.)
We thank the reviewer for the suggestion, but we cannot agree in this case provided, and as stated in the manuscript, there exists heteroscedasticity and serial correlation presented and identified in the data sample.
- Results should be more deeply explained and described. I have not seen the limitation, advantages and disadvantages of the Authors analysis. I could not confirm that these results are valid.
Our results are robust to the use of several indicators of financial inclusion and competitiveness, as well as to the use of an aggregate sample from the two countries. The conclusions section has been rewritten in accordance to accommodate all these explanations. So, thanks for the valuable observations.
I recommend to:
- The scientific problem of the paper (research) has to be corrected. The research question should be changed to convert to the hypothesis.
This has been made in the new version of the revised manuscript. Thanks.
- Should extend their considerations because they are not clear to all potential readers
This has been made in the new version of the revised manuscript. Thanks.
- More clearly, explain and emphasise the main concept, research problem, contribution to the science and hypothesis in comparison with existing literature
This has been made in the new version of the revised manuscript. Thanks.
- Do new analysis and build new panel regression models (fixed or random effects) and choose appropriate model’s statistics (e.g. Breusch-Pagan test; Durbin-Wu-Hausman test etc.)
Provided it was found heteroscedasticity and serial correlation problems, the most suitable option to use in methodological terms is the FGLS model guaranteeing robust results. Thanks for the suggestion.
- The conclusion should be extended. (limitation, what are the advantages and disadvantages of this analysis?)
This has been made in the new version of the revised manuscript. Thanks. We suggest, although, a new reading of the revised sections (identified in blue in the manuscript) provided some were very big to reproduce all here.
In general, if the Authors do major revisions - rebuild and complete their data analysis and aim, this study will be valuable.
We would like to thank the reviewer for the considerations, suggestions, and appreciations made regarding our article.

Round 2
Reviewer 1 Report
The article has a significant improvement and is satisfactory according to the journal.
The author needs to correct item 2 and add a clear background theory for the hypothesis.
Congratulations to the team of authors and hope you continue to succeed in the future.
Author Response
Review #1
The article has a significant improvement and is satisfactory according to the journal.
The author needs to correct item 2 and add a clear background theory for the hypothesis.
Congratulations to the team of authors and hope you continue to succeed in the future.
Answer: We thank the valuable comment of the reviewer about the present status of the work. There is no clear theory for the hypothesis as far as we are aware. We just wanted to stretch that few empirical studies approach in a specific way the relationship between financial regulation, inclusion, and competitiveness in the banking sector.

Reviewer 3 Report
I am satisfied with the corrects to the article; of course, in my opinion, you could do a little research, but the paper meets all the requirements for publication, and it is certainly interesting and valuable.
Author Response
Review #2
I am satisfied with the corrects to the article; of course, in my opinion, you could do a little research, but the paper meets all the requirements for publication, and it is certainly interesting and valuable.
Answer: We would like to thank the reviewer the valuable comments and suggestions.
